

# A 5-year study (2014–2018) of the relationship between coastal phytoplankton abundance and intertidal barnacle size along the Atlantic Canadian coast

Ricardo A. Scrosati and Julius A. Ellrich

Department of Biology, St. Francis Xavier University, Antigonish, Nova Scotia, Canada

## ABSTRACT

Benthic–pelagic coupling refers to the ecological relationships between benthic and pelagic environments. Studying such links is particularly useful to understand biological variation in intertidal organisms along marine coasts. Filter-feeding invertebrates are ecologically important on marine rocky shores, so they have often been used to investigate benthic–pelagic coupling. Most studies, however, have been conducted on eastern ocean boundaries. To evaluate benthic–pelagic coupling on a western ocean boundary, we conducted a 5-year study spanning 415 km of the Atlantic coast of Nova Scotia (Canada). We hypothesized that the summer size of intertidal barnacles (*Semibalanus balanoides*) recruited in the preceding spring would be positively related to the nearshore abundance (biomass) of phytoplankton, as phytoplankton constitutes food for the nauplius larvae and benthic stages of barnacles. Every year between 2014 and 2018, we measured summer barnacle size in clearings created before spring recruitment on the rocky substrate at eight wave-exposed locations along this coast. We then examined the annual relationships between barnacle size and chlorophyll-*a* concentration (Chl-*a*), a proxy for phytoplankton biomass. For every year and location, we used satellite data to calculate Chl-*a* averages for a period ranging from the early spring (when most barnacle larvae were in the water) to the summer (when barnacle size was measured after weeks of growth following spring benthic recruitment). The relationships were always positive, Chl-*a* explaining nearly half, or more, of the variation in barnacle size in four of the five studied years. These are remarkable results because they were based on a relatively limited number of locations (which often curtails statistical power) and point to the relevance of pelagic food supply to explain variation in intertidal barnacle size along this western ocean boundary coast.

## INTRODUCTION

Benthic–pelagic coupling refers to the ecological relationships that exist between benthic and pelagic environments (*Griffiths et al., 2017*). Recognition of such links has particularly facilitated progress in the field of intertidal ecology. For example, understanding how

Corresponding author
Ricardo A. Scrosati, rscrosat@stfx.ca

pelagic food supply and oceanographic features vary along coastlines frequently helps to predict, directly or indirectly, alongshore properties of intertidal species. Such studies, however, have overwhelmingly been conducted on eastern ocean boundaries (*Navarrete et al., 2005*; *Blanchette et al., 2008*; *Menge & Menge, 2013*; *Salant & Shanks, 2018*).

For western ocean boundaries, a question needing more research remains to what extent alongshore variation in intertidal species traits can be inferred from nearshore pelagic variables. On the SW Atlantic coast, for example, the recruitment of intertidal filter-feeders (barnacles and mussels) was recently found related to the abundance of phytoplankton (their main food source), wave exposure, and seawater temperature (*Arribas et al., 2014*; *Mazzuco et al., 2015*). On the NW Atlantic coast, surveys in the Gulf of Maine suggested that intertidal filter-feeder recruitment might be influenced by currents affecting larval supply (*Bryson, Trussell & Ewanchuk, 2014*). Larger-scale NW Atlantic surveys including sites on Canadian and American shores have found links between coastal phytoplankton abundance and intertidal barnacle recruitment (*Cole et al., 2011*) and between thermal stress during low tides and intertidal mussel abundance (*Tam & Scrosati, 2011*).

The Atlantic Canadian coast in Nova Scotia is well suited to study benthic–pelagic coupling, as it runs for some hundreds of km facing the open ocean. A study in 2014 revealed that the recruitment of intertidal barnacles and mussels in wave-exposed locations along this coast was positively related to pelagic food supply and, to a lesser degree, seawater temperature. In turn, recruitment of these filter-feeders was related to their abundance later in the year and, ultimately, to the abundance of their main predators (dogwhelks), suggesting bottom-up community regulation (*Scrosati & Ellrich, 2018*). While filter-feeder recruitment may predict predator abundance and even facilitation on other organisms (*Menge, 1976*), filter-feeder size is another important aspect of bottom-up regulation, as larger sizes represent more food for higher trophic levels (*Dunkin & Hughes, 1984*; *Carroll & Wethey, 1990*). Therefore, in this paper, we focus on barnacle size. Using field data for five consecutive years (2014–2018), we test the hypothesis that phytoplankton abundance (biomass) is positively related to intertidal barnacle size along this coast.

## MATERIALS AND METHODS

From 2014 to 2018, we collected data at eight intertidal locations spanning 415 km of the Atlantic coast of Nova Scotia (Fig. 1). For ease of interpretation, these locations are referred to as L1 to L8, from north to south (their names and coordinates are given in Table 1). They all have stable bedrock as substrate and are wave-exposed, as they face the open ocean directly. Daily maximum water velocity (an indication of wave exposure) measured in exposed intertidal habitats along this coast ranges between 6–12 m s$^{-1}$ (*Hunt & Scheibling, 2001*; *Scrosati & Heaven, 2007*; *Ellrich & Scrosati, 2017*). Using wave-exposed intertidal habitats to study benthic–pelagic coupling is particularly fitting because such places face the open ocean, which facilitates the identification of pelagic influences.

We measured the size of *Semibalanus balanoides*, which is the only intertidal barnacle species on this coast. For each location, we considered the intertidal range to be the vertical distance between chart datum (0 m in elevation, or lowest normal tide in Canada) and the

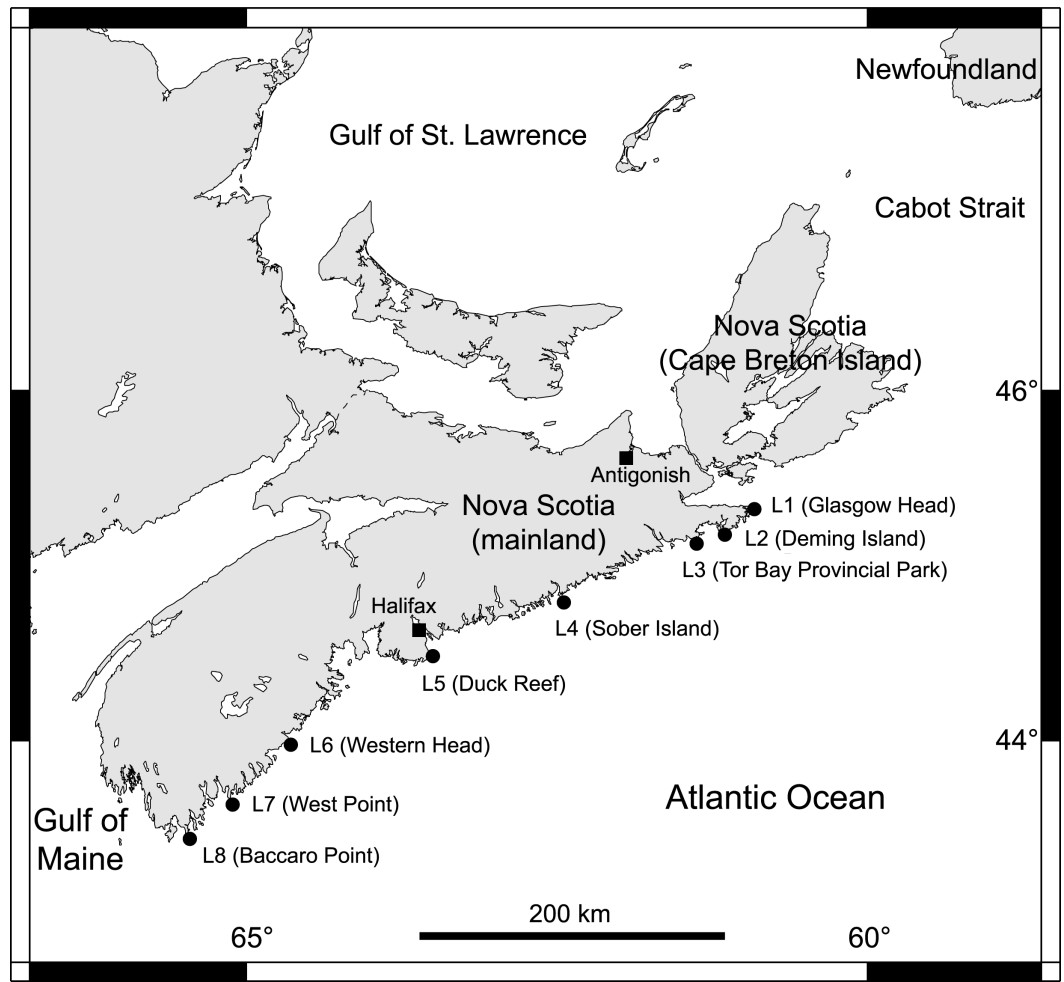

**Figure 1** **Map of the studied locations.** Map indicating the eight wave-exposed locations studied along the Atlantic coast of Nova Scotia, Canada.

highest elevation where sessile perennial organisms (coincidentally, *S. balanoides*) occurred on the substrate outside of crevices (*Scrosati & Heaven, 2007*). We divided the intertidal range by three and measured barnacle size just above the bottom boundary of the upper third of the intertidal range. As tidal amplitude increases from 1.8 m in L1 to 2.4 m in L8 (*Tide-Forecast, 2019*), this method allowed us to measure barnacle size along the coast at comparable elevations in terms of exposure to aerial conditions during low tides.

In Atlantic Canada, *Semibalanus balanoides* mates in autumn, broods in winter, and releases pelagic larvae in spring (*Bousfield, 1954*; *Crisp, 1968*; *Bouchard & Aiken, 2012*). Larvae settle in intertidal habitats and metamorphose into benthic recruits during May and June, which is thus considered to be the recruitment season (*Ellrich, Scrosati & Molis, 2015*). To measure barnacle size unaffected by other sessile species (*Beermann et al., 2013*), we made clearings (100-cm$^2$ quadrats) of the substrate in late April of each year along the same intertidal elevation (see above) at each location. Table 2 gives the number of

**Table 1  Information on locations.** Names and coordinates of the eight wave-exposed intertidal locations examined for this study and coordinates of the centre of the 4-km-x-4-km cells from which Chl-*a* data were extracted.

| Location code | Location name | Location coordinates | Cell centre coordinates |
|---|---|---|---|
| L1 | Glasgow Head | 45.3203, −60.9592 | 45.3125, −60.9791 |
| L2 | Deming Island | 45.2121, −61.1738 | 45.2292, −61.1875 |
| L3 | Tor Bay Provincial Park | 45.1823, −61.3553 | 45.1875, −61.3542 |
| L4 | Sober Island | 44.8223, −62.4573 | 44.8125, −62.4375 |
| L5 | Duck Reef | 44.4913, −63.5270 | 44.4797, −63.5208 |
| L6 | Western Head | 43.9896, −64.6607 | 43.9797, −64.6458 |
| L7 | West Point | 43.6533, −65.1309 | 43.6458, −65.1458 |
| L8 | Baccaro Point | 43.4496, −65.4697 | 43.4375, −65.4792 |

**Table 2  Survey dates and sample sizes.** Dates on which the quadrats were photographed. The number of available quadrats with barnacle size data is provided in parenthesis.

| Location | 2014 | 2015 | 2016 | 2017 | 2018 |
|---|---|---|---|---|---|
| L1 | 17 August (8) | 4 September (17) | 22 August (8) | 16 August (12) | 13 August (8) |
| L2 | 9 August (8) | 28 August (7) | 22 August (8) | 25 August (8) | 13 August (8) |
| L3 | 10 August (4) | 28 August (11) | 25 August (3) | 28 August (8) | 13 August (7) |
| L4 | 13 August (7) | 2 September (16) | 27 August (4) | 19 August (8) | 14 August (8) |
| L5 | 12 August (7) | 1 September (21) | 21 August (7) | 22 August (8) | 11 August (8) |
| L6 | 12 August (8) | 31 August (20) | 20 August (8) | 21 August (8) | 10 August (8) |
| L7 | 11 August (2) | 30 August (14) | 19 August (7) | 18 August (7) | 10 August (7) |
| L8 | 11 August (6) | 29 August (8) | 19 August (3) | 18 August (5) | 10 August (7) |

quadrats with barnacle size data for each location and year. Different quadrats were cleared each year by removing all pre-existing organisms from the substrate using a chisel and a metallic mesh scourer. We measured the size of the barnacles recruited therein as they looked in summer after growth (Table 2, Fig. 2). We determined barnacle size as the basal shell diameter measured along a straight line passing through the middle of the rostrum and the carina (*Chan et al., 2006*). To avoid influences of intraspecific crowding (*Bertness, 1989*) on our size data, we only measured barnacles that were not in contact with any neighbouring barnacles. Such organisms were common because, in summer, barnacles constituted almost the only macroscopic species in the quadrats and their density was not particularly high (Fig. 2). We measured size for a maximum of 10 such barnacles per quadrat (all barnacles if there were 10 or less in a quadrat and a random selection of 10 if there were more than 10 in a quadrat). For data analyses, we first calculated mean barnacle size for each quadrat and, then, averaged the corresponding quadrat means to generate a value of mean barnacle size per location and year.

To describe phytoplankton abundance, we used MODIS-Aqua satellite data on the concentration of chlorophyll-*a* in seawater (Chl-*a*, hereafter) for the 4-km-x-4-km cells that include our eight locations (*NASA, 2019a*). The coordinates of each cell are
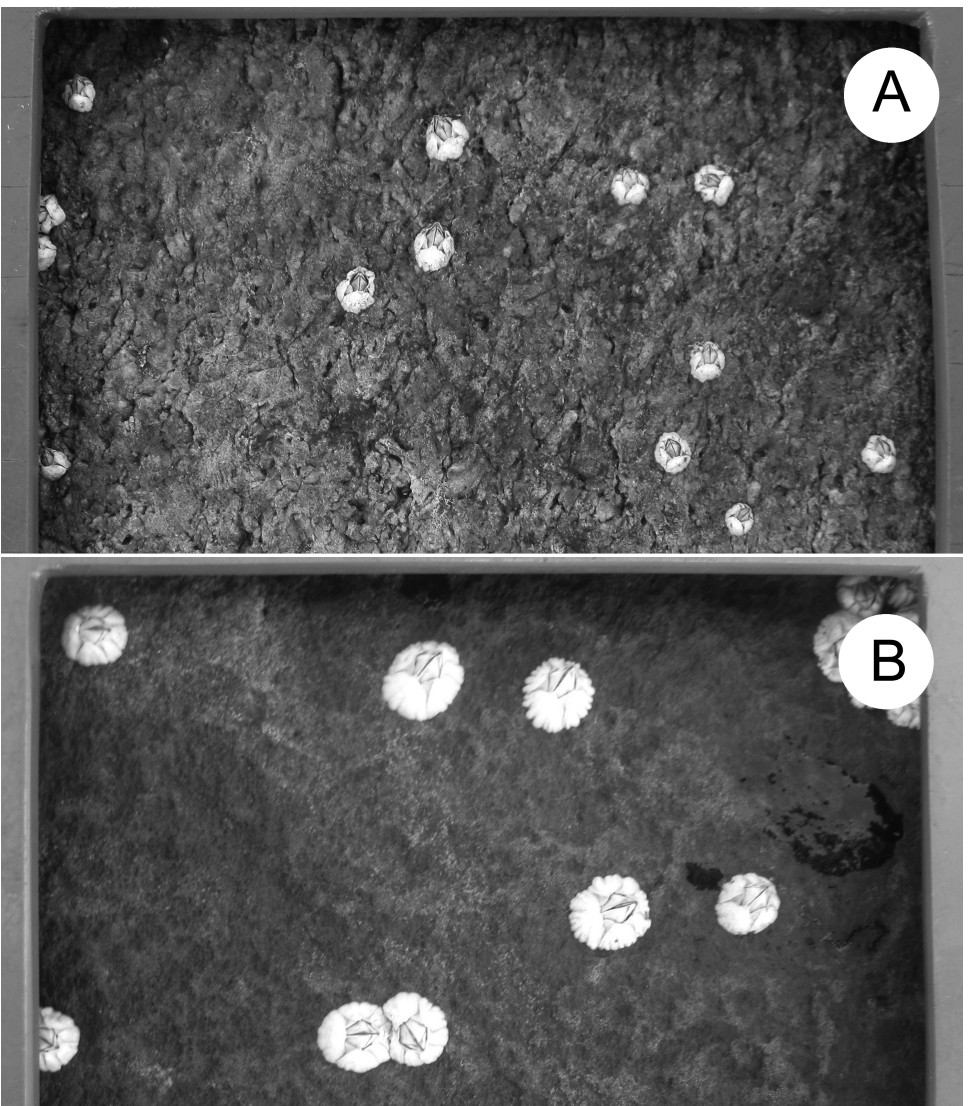

**Figure 2 Barnacle size.** Example of barnacle size differences between locations: (A) L1 and (B) L6. The frame bordering each photo belongs to the sampling quadrat. One full side of the quadrat (10 cm) is shown at the top of both pictures. The photos were taken by Ricardo A. Scrosati in August 2018.

stated in Table 1. Satellite Chl-*a* data are often used in intertidal ecology (*Navarrete et al., 2005*; *Burrows et al., 2010*; *Arribas et al., 2014*; *Mazzuco et al., 2015*; *Lara et al., 2016*) and are especially useful when studying hundreds of km of coastline for which in-situ phytoplankton data are lacking (*Legaard & Thomas, 2006*). For this study, satellite data should be appropriate because neighbouring locations are considerably more distant from one another than the data cell size (Fig. 1). To retrieve the Chl-*a* data, we first obtained mapped Chl-*a* data from the OceanColor database (*NASA, 2019a*). From these mapped data, we extracted the Chl-*a* values for our locations using the pixel extraction function of SeaDAS (*NASA, 2019b*) and the coordinates of our locations (Table 1). For each location

and year, we calculated the mean of all of the daily Chl-*a* values that were available from the beginning of April to the date when we measured barnacle size in summer (Table 2). Although barnacle recruitment occurs in May and June, we considered April Chl-*a* because of its possible effects on larval condition ultimately influencing benthic growth (*Barnes, 1956*; *Emlet & Sadro, 2006*). Specifically, the nauplius larvae of *S. balanoides* feed for 5–6 weeks in coastal waters before reaching the settling cyprid stage (*Bousfield, 1954*; *Drouin, Bourget & Tremblay, 2002*), and a recent study in our region concluded that most of the larvae that result in recruits are likely in the water in April (*Scrosati & Ellrich, 2016*). The Chl-*a* values between May and the summer dates when we measured barnacle size were used to represent pelagic food supply for the growing recruits.

For each year, we investigated the relationship between phytoplankton abundance and barnacle size by evaluating Pearson's correlation between the location means of Chl-*a* and size. As our hypothesis was directional (a positive association between both variables), we performed one-tailed tests of significance (*Quinn & Keough, 2002*). We also calculated the coefficient of determination for each year to evaluate the amount of variation in barnacle size that was statistically explained by Chl-*a*. We did the analyses with R version 3.5.1 (*R Core Team, 2018*). The data used for this study are available in figshare (doi: 10.6084/m9.figshare.7212446.v1).

## RESULTS

The observed relationships between Chl-*a* and barnacle size were always positive. The correlation coefficient was significant for 2014, 2016, and 2018 under a significance level of 0.05 and for 2015 under a less conservative significance level of 0.10 (Fig. 3). For 2017, the correlation was not significant, but still associated to a low *P* value ($P = 0.119$), suggesting a weak relationship that was hard to detect. As more data (more locations) for 2017 were unavailable, we excluded the southernmost location (L8) from that year's dataset because L8 then exhibited the lowest mean barnacle size (<0.3 cm) for the entire dataset used for this study. This modification yielded a significant correlation coefficient (Fig. 3), indicating that a positive size–Chl-*a* relationship also existed for 2017, although for a more limited geographic extent that excluded the southern end of the studied coast. Chl-*a* explained 49% of the variation in barnacle size in 2014, 32% in 2015, 62% in 2016, 47% in 2017 (excluding L8), and 47% in 2018. Each year, barnacle size was highest at the same two neighbouring southern locations (L6 and L7). Both such locations also exhibited a higher average Chl-*a* than the average for the other six locations each year (Fig. 3).

## DISCUSSION

This 5-year study along the Nova Scotia coast has revealed that the summer size of intertidal barnacles recruited in the spring was positively related to the mean phytoplankton abundance for the preceding months. This outcome is remarkable because the five examined correlations were based on data for just eight locations (seven in 2017), a limited number that, by curbing statistical power, often prevents field studies from detecting patterns in ecology. Surveying more wave-exposed locations was not feasible because of safety concerns

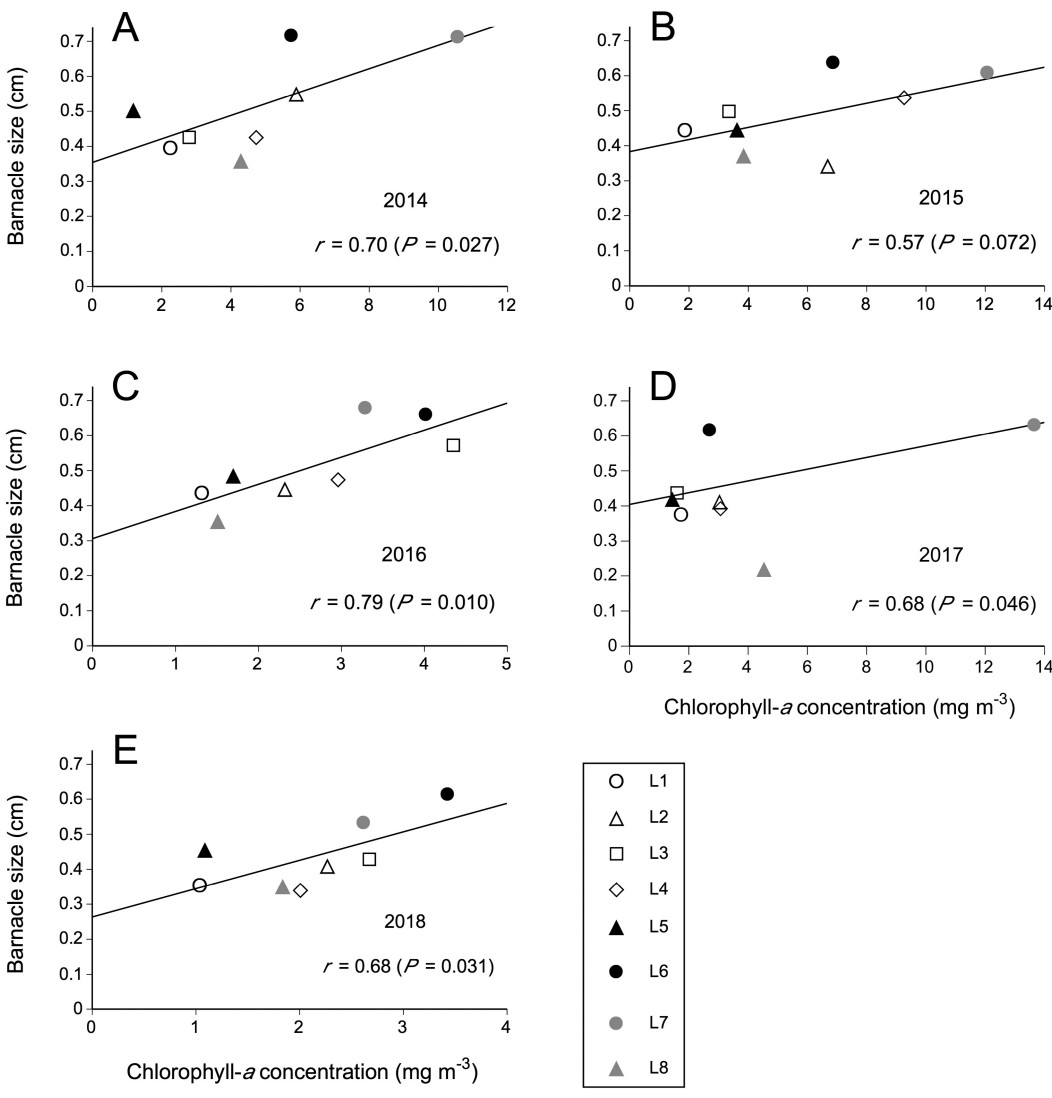

**Figure 3** **Relationships between coastal phytoplankton abundance (chlorophyll-*a* concentration) and intertidal barnacle size in (A) 2014, (B) 2015, (C) 2016, (D) 2017, and (E) 2018.** The correlation and functional relationship shown for 2017 were calculated without including L8 (see Results for rationale); the data point for L8 is nonetheless shown in this figure simply for visual reference.

(dangerous site access or lack of roads) and the need to sample all locations within a few days of difference every year. Thus, the observed correlations highlight the relevance of Chl-*a* to statistically explain the alongshore variation in barnacle size. Interestingly, in four of the five studied years (2014, 2016, 2017 –excluding L8–, and 2018), Chl-*a* explained nearly half, or more, of such variation.

These results are likely explained by the fact that phytoplankton is the main food source for barnacle nauplius larvae and benthic stages (*Anderson, 1994*; *Jarrett, 2003*; *Gyory, Pineda & Solow, 2013*). This consideration bears special relevance in light of the alongshore variation of Chl-*a* (Fig. 3), since, for the 5 years of this study, the mean annual coefficient of

variation for Chl-*a* for our locations was 61%. Given the temporal intra-annual resolution of the Chl-*a* data, however, it is not possible to ascertain if the purported role of phytoplankton may have differed depending on the developmental stage of barnacles (from pelagic larvae to the successive benthic stages until size was measured). Thus, this study should be best viewed as broad evidence revealing benthic–pelagic coupling on the NW Atlantic coast using variables not previously examined together for this system (*Bryson, Trussell & Ewanchuk, 2014*; *Scrosati & Ellrich, 2018*). Ultimately, this study suggests that a spatial association between phytoplankton abundance and filter-feeder growth can occur on a western ocean boundary, adding to similar findings reported for eastern ocean boundaries (*Menge et al., 1997*; *Menge et al., 2003*).

A consideration about satellite Chl-*a* data is worth making at this point. In recent years, several studies on intertidal ecology that needed information on coastal phytoplankton used satellite Chl-*a* data as a proxy (*Navarrete et al., 2005*; *Burrows et al., 2010*; *Arribas et al., 2014*; *Mazzuco et al., 2015*; *Lara et al., 2016*). Although such data are useful for coasts for which in-situ phytoplankton data are unavailable (*Legaard & Thomas, 2006*), the fact remains that a proxy is being used. For our study, that was the only option. Because our goal was to evaluate statistical relationships between summer barnacle size and average phytoplankton abundance from spring to summer, having in-situ phytoplankton data would have required frequent seawater samplings during those months at each of our eight locations for the five years of the study. However, that was not possible for logistical reasons. Oceanographic research is continuously improving the algorithms to accurately infer coastal phytoplankton abundance from satellite data (*Bellacicco et al., 2016*; *Yang et al., 2018*). Thus, future studies could re-examine our hypothesis as further improvements are made on that line. If the logistics to generate in-situ phytoplankton data for the required spatial and temporal scales can be solved, our hypothesis could also be examined using direct phytoplankton data. In any case, given the consistently positive relationships between barnacle size and Chl-*a* found for our coast, that our wave-exposed locations fully face open oceanic waters, that no major rivers occur along the studied coast, that human population density is very low along this coast, and that pelagic food supply enhances benthic filter-feeder growth (*Menge et al., 1997*; *Menge et al., 2003*), it seems reasonable to expect positive relationships under those improved approaches as well.

From the results of the present study, an emerging question of interest is what may cause variation in phytoplankton abundance along the Nova Scotia coast. The intermittent upwelling hypothesis (IUH; *Menge & Menge, 2013*) refers to possible mechanisms. The IUH considers that frequent wind-driven upwelling would limit coastal phytoplankton abundance because upwelled nutrients (necessary for phytoplankton development) would be taken offshore before nearshore blooms can occur. Frequent downwelling would also limit coastal phytoplankton abundance by driving nutrient-poor surface waters to the coast. Intermittent upwelling, however, would allow upwelled nutrients to remain near the coast long enough for phytoplankton to bloom, thus favouring the growth of intertidal filter-feeders (*Menge & Menge, 2013*). Wind-driven upwelling has been reported for the Atlantic coast of Nova Scotia (*Petrie, Topliss & Wright, 1987*; *Shan et al., 2016*), making the IUH worth testing for this coast. In particular, a study in 1984 reported coastal cooling

between June and July near L6 and L7, while seawater temperature increased for the same period of time near our northern locations (*Petrie, Topliss & Wright, 1987*). The localized upwelling on the Atlantic coast of Nova Scotia is probably intermittent, less frequent and intense than on heavy-upwelling coasts like that of California. Therefore, the fact that mean Chl-*a* for L6 and L7 was higher than for the other locations every year suggests that the IUH might help to understand alongshore variation in phytoplankton abundance and, ultimately, intertidal barnacle growth on our coast. Alternative analytic approaches, however, have found no support for the mechanisms underlying the IUH, suggesting that surf zone width and tidally generated internal waves better explain changes in coastal phytoplankton abundance (*Salant & Shanks, 2018*; *Shanks & Morgan, 2018*). At present, this topic is undergoing an active debate (*Menge & Menge, 2019*; *Shanks & Morgan, 2019*). Whatever the causes of Chl-*a* variation along the Nova Scotia coast, it is likely that some combination of oceanographic properties is involved. These properties (upwelling, surf zone width, internal waves, etc.) could change differently every year along the coast, thus generating complex scenarios worth investigating from an oceanographic standpoint.

# CONCLUSIONS

Based on data collected over five years (2014–2018) for locations spanning 415 km of the Atlantic Canadian coast, this study reveals a persistent relationship between the summer size of intertidal barnacles recruited in the preceding spring and the nearshore abundance of phytoplankton (food for nauplius larvae and benthic stages of barnacles). Phytoplankton abundance, measured through satellite Chl-*a* data, explained nearly half, or more, of the alongshore variation in barnacle size in four of the five studied years. These are remarkable results because they were based on a relatively limited number of locations and point to the relevance of pelagic food supply to explain variation in intertidal barnacle size along this western ocean boundary coast.

# ACKNOWLEDGEMENTS

We thank Carmen Denfeld, Willy Petzold, and Maike Willers for field assistance and two anonymous reviewers for constructive comments on an earlier version of this paper.

## Funding

This project was funded by grants awarded to Ricardo A. Scrosati by the Natural Sciences and Engineering Research Council of Canada (Discovery Grant #311624), the Canada Foundation for Innovation (Leaders Opportunity Grant #202034), the Canada Research Chairs program (CRC Grant #210283) and by a postdoctoral fellowship (#91617093) awarded to Julius A. Ellrich by the German Academic Exchange Service (DAAD). The funders had no role in study design, data collection and analysis, decision to publish, or preparation of the manuscript.

## Grant Disclosures

The following grant information was disclosed by the authors:

Natural Sciences and Engineering Research Council of Canada (Discovery Grant): # 311624.

Canada Foundation for Innovation (Leaders Opportunity Grant): #202034.

Canada Research Chairs program (CRC Grant ): #210283.

German Academic Exchange Service (DAAD): #91617093.

## Competing Interests

The authors declare there are no competing interests.

## Author Contributions

- Ricardo A. Scrosati conceived and designed the experiments, performed the experiments, analyzed the data, contributed reagents/materials/analysis tools, prepared figures and/or tables, authored or reviewed drafts of the paper, approved the final draft.
- Julius A. Ellrich performed the experiments, analyzed the data, approved the final draft.

## Data Availability

The raw data are available from figshare: Scrosati, Ricardo A.; Ellrich, Julius A. (2018): Chlorophyll-a and barnacle size data. figshare. Dataset. https://doi.org/10.6084/m9. figshare.7212446.v1.

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
