# Peer review of "A 5-year study (2014–2018) of the relationship between coastal phytoplankton abundance and intertidal barnacle size along the Atlantic Canadian coast"

_PeerJ, doi:10.7717/peerj.6892_

## Round 0.1 · original submission · Major Revisions

Two reviewers examined your manuscript. Both agree that the study is well written and contribute relevant results. The reviewers, however, identify some limitations of the study. Both reviewers have concerns with the use of satellite chl a concentrations, and its variation. Reviewer #2 suggested the use additional satellite data (e.g. temperature) to help explain the variation. They also suggest changes and revisions in the abstract, methods results and discussion.

Reviewer 1 ·

Basic reporting

No comment.

Experimental design

No comment.

Validity of the findings

No comment.

Additional comments

1. General comments
In this paper, the authors report on a 5-year study (2014-2018) on the relationship between coastal phytoplankton biomass and intertidal barnacle size along the Atlantic Canadian coast. ” To do so, they used satellite data to document chl a concentrations and they measured the size of intertidal barnacles over this period at eight locations along 415 km of the Atlantic coast of Nova Scotia. For almost every year, the authors found a positive relationship between the summer barnacle size and chl a concentrations for the months before summer. This study adds on current research in the field benthic-pelagic coupling along marine intertidal coasts. The scientific value of the paper could be considered as high since it helps explaining the inter-individual variation in barnacle size along the western boundary of the Atlantic. The topic meets the criteria laid down for publication in PeerJ. The manuscript is generally well written and could be accepted after some revision (see specific comments).

2. Specific comments
Abstract: Although the content of the abstract is informative, it should present the working hypothesis that appears in the Introduction.

Introduction: There is little published information available on benthic-pelagic coupling along the western boundary of the Atlantic. As a result, the results of the present study could be valuable. This is somewhat well explained. The objectives are relevant and strengthen the content of the paper.
Lines 47-50: A reference is needed here. Also, the fact that the temperature is not taken into account in the present study should be justified.
Line 58: I would suggest to use the term biomass rather than abundance to relate phytoplankton chl a concentrations to barnacle size.

The Material and methods are adequate for the problem tackled and the analyses seem to be justified for the data set. Overall, this section is well-written.
Results: Overall, the presentation of results is fine. Some clarifications, however, are needed on two aspects.
Lines 122-128: How do the authors explain the lack of significant correlation between chl a concentrations and barnacle size in 2017? Is there an effect of the water temperature?

Presentation of discussion and conclusion: The discussion is well-written and the answers to the objectives presented in the introduction are reached and convincing. However, I wonder of it is necessary to have a long paragraph discussing about the causes of the variation of chl a concentrations along the coast of Nova Scotia. The end of the discussion is interesting but speculative.
Lines 144-146: It is written that Chl a concentrations explain half or more of the alongshore variation in barnacle size. Do the authors have any idea of other potential factors explaining this variation?

Reviewer 2 ·

Basic reporting

I have perception that document is result like cut and paste derived secondarily from past publication not a new independent "stand alone" work, not establishing new ground integration satellite chlorophyll-a data with in-situ sampling has a sense of being "tacked on" to establish basis for paper. However, I like idea and has great potential if in the analysis use additional available satellite data (thermal> gulf stream?, visual photos > discern wave action, rip tide, calm zones of inaccessible areas to compare to areas he has access to. See if something pops out of the noise), NOAA currents, wind charts, bathymetric data. Needs to study and utilize existing public databases to solidify work The author needs to expand his vision on pertinent investigative avenues and oceanographic viewpoint. Oceanographic features important but not considered.

A weakness of the work, that has to be considered, lies in the use of chlorophyll data derived from satellite sensors is problematic, its use for coastal waters can be skewed, in addition to assuming that this data is an indicator of phytoplankton abundance.

Though out document makes a lot of general statements, not specific point, then cites literature where information contained. Should be giving specific information from literature pertinent to argument and integrate into current work with citation. Example: Line 157-159 obvious, because barnacles are on the rocks of Nova Scotia and they are filter feeders. How is a spatial connection established that “can occur”, adding to the relationships previously reported for eastern ocean boundaries (Menge et al., 1997, 2003). Reader made do a lot of work to see if what said is relevant. Should give relevant specific information, supporting work and citation

Experimental design

Original primary research within Aims and Scope of the journal. Research question is well defined

The methods were not written in enough detail. some specific details are listed:

Was there a specific reason for the selection of 100 sq. cm. area? In your methodology isn’t clear did you use same exact quadrant each year cleaning off last year’s growth or establish new quadrant in same area?

Not clearly stated that you are establishing your transects along coast and have more than one quadrant at each site until look at table 2, available quadrants in parenthesis, at each station

Line 17: importance of site exposure to wave action

Line 93: Your usage of “random” selection. In one photo A I see 10 candidates and in the other photo B, 7 candidates. What is your criteria for making “random”, clarify please.

It is suggested to include more details of the chlorophyll data, such as the coordinates of the area considered for the extraction of the data and their temporal resolution.

Validity of the findings

Assuming equivalence between coastal phytoplankton abundance and chlorophyll-a concentration is not correct.

Phytoplankton abundance and/or biomass does not covary with chlorophyll over the entire seasonal cycle. Chlorophyll cell concentration changes as response in nutrients and light conditions. Thus Chlorophyll is a poor indicator of the phytoplankton abundance or biomass variations. I suggest the author review the publication of Bellacicco et al 2016 and references there in (http://dx.doi.org/10.1016/j.rse.2016.08.004)

Lines 97-102 Chlorophyll-a concentration from coastal waters color satellite is affected by other water constituents, such as non-phytoplankton particles and colored dissolved organic matter. In what way are you considering this? I suggest reading Yang et al., 2018. Remote Sens. 2018, 10, 1335; doi:10.3390/rs10091335


Article based on satellite data, but very little available data utilized.

David (1938) recommends the use of Confidence Intervals for Pearson correlations only if n ≥ 25. Why didn’t author put all your raw data (total barnacle size data) instead of average in analysis? This would make his analysis more robust in supporting his results.

Line 127-128: Why exclude data? The reason is not clear.

David, F.N. (1938). Tables of the ordinates and probability integral of the distribution of the correlation coefficient in small samples. Cambridge: Cambridge University Press.

Bonett and Wright (2000), Sample size requirements for estimating Pearson, Kendall and Spearman correlations, Psychometrika, 65(1)

Additional comments

Why are filters feeders important along rocky shores? Important or ubiquitous?
Most likely in water larvae released to coincide w/ spring algal bloom. How does barnacle know algal bloom occurring? Type of Ecological signal? Phytoplankton concentration, Temp., etc. Article’s theme benthic-pelagic coupling ecological relation should be some type of environmental/ ecological signal. If relationship hasn’t been found should be stated reference cited (Scrosati & Ellrich, 2016)
Says currents could hinder recruitment by moving away from intertidal zones. No bathymetric, current data or wind chart data? Station # 8 close to Bay of Fundy tidal bore influence no mention of mechanical resuspension of sediments and nutrients and dispersive zone of influence. Turbidity
No mention of effect of Gulf Stream pinched off gyres or tributaries etc. satellite photographs thermal imagining
Line 22: chl-a explains nearly half of growth what’s other part
Line 51-57 relevance to article’s topic plankton abundance vs. growth
Change “were done on eastern ocean boundaries” by “have been conducted”
Could full moon be part of release signal, tides?
Line 139-141: “This outcome is remarkable because the five examined correlations were based on data for just eight locations (or seven in 2017), a limited number that, by curbing statistical power, often prevents field studies from detecting patterns in ecology.” Why not restate idea? Awkward phraseology
Line 162-164: upwelling move nutrients away from shore?? Explain add citation
Are there another local nutrients sources?
Currents transporting barnacle larvae along shore?
Could there be physical topographical/ bathymetric features that make a favorable microclimatic area?
Line 141-142 Safety concerns or lack of access renders the majority of coastline unknown. He has barnacle size data and chlorophyll data at eight sites, has satellite data for entire coastline.

---

## Round 0.2 · accepted · Accept

The authors have resolved the concerns on clarity and limitations identified by the reviewers.

# Reviewer 2 ·

Basic reporting

The submission defined the research question. The authors' clarifications are satisfactory to establish a sequential chain of a research for the relationship between coastal chlorophyll a and intertidal barnacle size.

Experimental design

No comment

Validity of the findings

No comment

Additional comments

After a careful revision of the manuscript "5-year study (2014-2018) on the relationship between coastal phytoplankton biomass and intertidal barnacle size along the Atlantic Canadian coast", the authors' clarifications and supplemental information improved the document to a satisfactory level. It is generally well written and could be accepted for publication in PeerJ.